# Factors Affecting Abdominal Obesity: Analyzing National Data

**DOI:** 10.3390/healthcare12080827

**Published:** 2024-04-14

**Authors:** Gwihyun Kim, Hyekyung Woo, Young-A Ji

**Affiliations:** 1Department of Health Administration, Kyungin Women’s University, Incheon 21041, Republic of Korea; adelagh@daum.net; 2Department of Health Administration, Kongju National University, Gongju 32588, Republic of Korea; 3Department of Medical Education, College of Medicine, Gyeongsang National University, Jinju 52727, Republic of Korea

**Keywords:** abdominal obesity, BMI, obesity, sleep time, subjective health status

## Abstract

The purpose of this study is to understand the factors affecting abdominal obesity. A secondary data analysis was conducted to analyze 5262 individuals’ data from the 2020 Korea National Health and Nutrition Examination Survey. The prevalence of obesity was slightly higher in men than women, while abdominal obesity was more prevalent in women. A higher correlation with obesity was observed in young and middle-aged individuals, unmarried individuals, urban residents, those with good subjective health, low-stress perception, moderate alcohol consumption, nonsmokers, regular aerobic exercisers, and those getting more than seven hours of sleep. In contrast, middle-aged and elderly individuals, married individuals, rural residents, those with an elementary school or lower education level, those with low-to-moderate income, those with fair or poor subjective health, high stress perception, nondrinkers, smokers, nonregular aerobic exercisers, and those getting less than seven hours of sleep had a higher correlation with abdominal obesity. Health education suggests that everyone should maintain healthy lifestyle habits, such as getting sufficient sleep, exercise, smoking cessation, and moderate drinking. Specifically, diverse health management support focusing on population groups with demographic factors related to the risk of obesity and abdominal obesity is necessary.

## 1. Introduction

### Significance and Objectives of the Study

Obesity is a global health problem with an increasing risk in all nationality, gender, and age groups. In some countries, more than 50% of adults are reported to be obese [1]; thus, obesity is a significant health risk for modern individuals. The World Health Organization (WHO) has classified obesity as a disease and identified it as a major contributing factor to colorectal and ovarian cancers [2,3]. Particularly, abdominal obesity is closely associated with increased risk of cardiovascular diseases, diabetes, and fatty liver diseases [4]. A sedentary lifestyle with low physical activity and high energy intake can lead to fat accumulation in the abdomen, increasing the likelihood of abdominal obesity.

According to WHO data from 2018, the global obese population has increased nearly three-fold compared to 1975, with over 1.9 billion people estimated to be overweight and more than 650 million estimated to be obese in 2016 [2]. CDC data from 2020 reported an obesity prevalence of 39.8% among the US adult population in 2015–2016, totaling approximately 93.3 million adults, while Finkelstein et al. estimated that, by 2030, 51% of the US population would be obese [5]. Ward et al. predicted that, by 2030, one in two adults would become obese [6]. In a nationwide survey in South Korea in 2020, 48.0% of adult men and 27.3% of women aged 19 and above were classified as obese based on body mass index (BMI) criteria [7]. Comparing this with data from 2015, which estimated obesity rates at 39.7% for adult men and 26.0% for [8], it is evident that obesity prevalence is increasing in both genders, especially with a significant increase in obesity rates among men.

Obesity is influenced by individual health characteristics, behavioral patterns based on personal interests, and socioeconomic factors. To initiate health-promoting actions, individuals must recognize the need for healthy behaviors and perceive the necessity for obesity prevention [9]. However, the obesity rate remained relatively stable around 34% after an increase from 31.7% in 2007 to 33.2% in 2015, then reaching 33.8% in 2019. However, there was a significant increase to 38.3% in 2020, and, as of 2022, it has been maintained at 37.2% [10]. A study by Nora et al. (2023) demonstrated an increase in obesity-promoting dietary behaviors and food choices during the COVID-19 pandemic. Research by Nour and Altintas (2023), which investigated obesity determinants during the pandemic across 22 countries, indicated that physical inactivity and poor dietary habits were contributing factors [11].

In this study, we aimed to identify the factors influencing obesity and abdominal obesity using data from the Korea National Health and Nutrition Examination Survey, a large-scale representative sample of the Korean population. We examined the relationship between socioeconomic and health behavior factors defined by the 4th Comprehensive Plan for National Health Promotion (2015–2020), including smoking, alcohol consumption, physical activity and stress, subjective health perception, sleep duration, depression, obesity, and abdominal obesity. The goal was to provide basic evidence for prevention and management strategies for the obese and abdominal obesity populations, taking into account different approaches and priorities.

## 2. Research Methods and Content

The methods section is structured into three main components: 1. Study Population, 2. Research Tools, and 3. Analysis Method, and below are the detailed aspects for each item to clearly describe them.

### 2.1. Study Population

This study utilized data from a nationwide health and nutrition survey conducted by the Ministry of Health and Welfare and the Korea Disease Control and Prevention Agency. Since infants and adolescents are both in a growth period and a school age period, the reasons for obesity will be different from those of adults, and the intervention methods must also be different, so this study targeted adults. Data collection included household confirmations, health examinations, and nutrition surveys. Specifically, the examination survey, conducted using methods such as direct measurement, observation, and sample analysis, provided reliable BMI data obtained through weight and height measurements. This study focused on data from the 8th round of the 2nd National Health and Nutrition Examination Survey conducted in 2020. After excluding individuals under 20 years of age, pregnant and lactating women, and those with missing values for variables used in the study (such as BMI and sleep duration), the study population comprised 2390 men (45.4%) and 2872 women (54.6%), totaling 5262 individuals.

### 2.2. Research Tools

(1).Dependent Variables

Although BMI has long been used as an indicator of obesity, recent approaches have also incorporated waist circumference as a criterion for diagnosing abdominal obesity. In this study, dependent variables related to obesity and abdominal obesity were created using data collected from weight, height, and waist circumference measurements during the examination survey. BMI was calculated using the formula (weight (kg)/height (m^2^)), and obesity and abdominal obesity were classified according to the Korean Society for the Study of Obesity’s Obesity Treatment Guidelines (2022). Obesity was defined as BMI equal to or greater than 25 kg/m^2^, and abdominal obesity was defined using waist circumference measurements, with 90 cm or more for men and 85 cm or more for women [12].

(2).Independent Variables

Socioeconomic characteristics (gender, age, marital status, household income, and education level) and health-related characteristics (perceived stress, sleep time, depression, smoking, drinking, subjective health perception, and aerobic exercise) were selected as independent variables.

Among socioeconomic characteristics, adults aged 20 or older were classified into three groups (young, middle-aged, and senior) at 20-year intervals. Household income was categorized into four quartiles of household income. Household income was classified into quartiles of household income according to the table of sample household income quintiles in the National Health and Nutrition Examination Survey data. Marital status was categorized as married and single, and educational background was categorized as elementary school graduation or less, middle school graduation, high school graduation, and college graduation or higher.

Based on the 4th National Health Promotion Plan (2015–2020), smoking, drinking, physical activity, cognitive stress, sleep time, depression, and subjective factors were added. Through a literature review, sleep time, which is considered an important variable in health behavior, was divided into less than 7 h and more than 7 [13,14]. Subjective health status, which had been classified into 5 categories (bad to very good) in the National Health and Nutrition Examination Survey data, was reclassified into 3 categories: good, average, and poor. Depression was classified as present only if diagnosed by a physician. The aerobic exercise practice rate was classified based on the periods of moderate-intensity physical activity, high-intensity physical activity, moderate-intensity physical activity, and high-intensity physical activity mixed according to the National Health and Nutrition Examination Survey data analysis guidelines.

### 2.3. Analysis Method

We analyzed data from the National Health and Nutrition Examination Survey conducted on the entire Korean population since 2020. The sample design for the National Health and Nutrition Examination Survey is a two-stage stratified cluster sampling design, and it is recommended that data reflecting this complex sample design be analyzed [7,15]. In accordance with the guidelines for using raw data from the National Health and Nutrition Examination Survey, weighted variables were applied using examination survey weights (variable name wt_itvex) for health surveys (health interviews/health behaviors) to conduct the analysis.

A cross-analysis was performed to examine the association between socioeconomic factors (gender, age, marital status, household income, residential area, and education level) and health-related characteristics (subjective health perception, sleep duration, perceived stress, depression, alcohol consumption, smoking, and aerobic exercise) and obesity and abdominal obesity among the study population. Weights were applied to compensate for population structure and missing values. Furthermore, binary logistic regression analysis was conducted to identify factors influencing obesity and abdominal obesity based on socioeconomic and health-related characteristics. The collected data were analyzed using SPSS software, version 23.0, and the significance level for statistical hypothesis testing was set at 0.05.

## 3. Research Results

### 3.1. Socioeconomic Characteristics and Obesity

The total study population consisted of 5262 individuals, with an obesity rate (BMI ≥ 25 kg/m^2^) of 38.3% (2014 individuals; 1102 men and 912 women). The rate of abdominal obesity was 39.1% (2057 individuals; 1088 men and 969 women). Even when comparing the same study population, the proportion of abdominal obesity was higher than that of obesity, with slightly higher rates of obesity and abdominal obesity observed in men and women, respectively. Significant age-related differences were observed, with higher obesity rates in middle-aged (41.3%) and young (33.3%) individuals, and higher abdominal obesity rates in older (48.5%) and middle-aged (40.6%) individuals.

Regarding marital status, the prevalence of abdominal obesity (80.1%, obesity 75.9%) was higher in married individuals than in unmarried individuals (24.1%, abdominal obesity 19.9%). In terms of residential area, urban residents showed a higher obesity rate (84.3%), while rural residents exhibited a higher rate of abdominal obesity (16.8%). The results based on household income, categorized into four quartiles, and education level showed higher rates of obesity and abdominal obesity for those with college education or above (42.9%, 38.1%), higher income (33.2%, 32.3%), and upper-middle income (30.9%, 28.2%). Conversely, individuals with lower education (elementary school graduate or below: 11.5%, 15.4%), lower-middle income (22.1%, 23.4%), and low income (13.7%, 16.1%) had a higher proportion of abdominal obesity than those with obesity (Table 1).

### 3.2. Obesity According to Health Behavior Characteristics

In terms of the results of comparing obesity and abdominal obesity according to health behavior characteristics, individuals reporting good subjective health status showed a higher obesity rate (28.3%; abdominal obesity, 24.7%). However, when the subjective health status was average or poor, the rate of abdominal obesity was higher (average, 51.9%; poor, 23.4%) than that of obesity (average, 50.5%; poor, 21.2%). Those who perceived high stress levels exhibited a higher rate of abdominal obesity (69.4%; abdominal obesity, 70.2%), whereas those who perceived low stress levels showed a higher rate of obesity (30.6%; abdominal obesity, 29.8%).

The group that consumed alcohol had a higher obesity rate (92.3%; abdominal obesity, 90.5%), whereas the group that did not consume alcohol had a higher rate of abdominal obesity (7.7%; abdominal obesity, 9.5%). In contrast, the smoking group had a higher rate of abdominal obesity (42.7% vs. 43.4%), whereas the nonsmoking group exhibited a higher rate of obesity (57.3% vs. 56.6%).

Engaging in aerobic exercise resulted in a higher obesity rate than abdominal obesity (46.2% vs. 41.9%), whereas those not practicing aerobic exercise showed a higher rate of abdominal obesity (53.8% vs. 58.1%). When analyzing sleep duration, individuals sleeping seven hours or more exhibited a higher obesity rate than those with abdominal obesity (60.3%; abdominal obesity 58.0%). Conversely, those sleeping less than seven hours showed a higher rate of abdominal obesity (obesity, 39.7%; abdominal obesity, 42.0%). Additionally, in the analysis of average sleep duration (7.00 ± 1.38), both the obesity (6.91 ± 1.37) and abdominal obesity (6.83 ± 1.40) groups had shorter average sleep durations. Those without obesity (7.06 ± 1.37) and abdominal obesity (7.11 ± 1.35) had longer average sleep durations than the overall mean sleep duration (Table 2).

### 3.3. Factors Influencing Obesity and Abdominal Obesity According to Gender

Since there were gender differences in obesity and abdominal obesity, logistic regression analysis was conducted for the overall target group as well as for subgroups divided into men and women to identify the factors influencing obesity and abdominal obesity. The results are summarized in Table 3.

All the variables significantly influenced obesity and abdominal obesity. Compared with women, the likelihood of obesity in men increased by 2.318 times, while the likelihood of abdominal obesity increased by 1.857 times (*p* < 0.01). Furthermore, in comparison with the elderly, obesity was 1.114 times higher in young men and 1.477 times higher in young women, while it was 0.953 times higher in middle-aged men and 1.660 times higher in middle-aged women. Abdominal obesity was 0.914 times lower in young men, 0.965 times lower in young women, 0.895 times lower in middle-aged men, and 1.379 times higher in middle-aged women, indicating that obesity and abdominal obesity were higher in middle-aged women than in men (*p* < 0.01).

Married individuals showed a probability increase of 1.273 times for obesity and 1.051 times for abdominal obesity compared with unmarried individuals. Both obesity and abdominal obesity were more prevalent in men than in women (*p* < 0.01). Urban residents had a 1.137 times higher probability of obesity and a 1.163 times higher probability of abdominal obesity than rural residents. In addition, obesity (1.830 times) and abdominal obesity (1.833 times) were more common in women than in men, whereas obesity (1.112 times) and abdominal obesity (1.130 times) were more common in men than in women (*p* < 0.01).

In the lower-income group, obesity increased 1.053 times and abdominal obesity increased 0.783 times, but obesity increased with higher income. Abdominal obesity was higher in middle-to lower-income groups (*p* < 0.01). Regarding educational level, as education improved, both obesity and abdominal obesity increased. The increases in obesity and abdominal obesity in women were significantly greater than in men. The obesity levels in men and women with elementary school graduation or below were 0.745 and 0.274, respectively, whereas the abdominal obesity values were 0.785 and 0.506, respectively. For men and women with a middle school education, obesity was 0.725 and 1.418 times, respectively, and abdominal obesity was 0.870 and 2.220 times, respectively. For men and women with a high school education, obesity was 0.823 and 1.627 times, respectively, and abdominal obesity was 0.953 and 1.668 times, respectively, indicating a larger increase in women’s obesity and abdominal obesity (*p* < 0.01).

According to sleep duration, those sleeping less than seven hours had a 1.310 times higher probability of obesity and a 0.919 times higher probability of abdominal obesity than those sleeping seven hours or more. Men had a 1.304 times higher probability of obesity and a 1.076 times higher probability of abdominal obesity, whereas women had a 1.538 times higher probability of obesity and a 1.478 times higher probability of abdominal obesity (*p* < 0.01). Compared with those with good subjective health, those with intermediate subjective health had a 0.702 times higher probability of obesity and a 0.522 times higher probability of abdominal obesity. When feeling bad, the probabilities of obesity and abdominal obesity increased by 0.723 and 0.662 times, respectively. The increases in obesity and abdominal obesity in women were approximately twice those in men (*p* < 0.01). Regarding current depression, compared to not being depressed, the probability of obesity increased by 1.217 times and that of abdominal obesity by 1.032 times. Women showed an increase in obesity and abdominal obesity approximately twice as much as men (*p* < 0.01).

When the perceived stress rate was high, the probability of obesity increased 1.020 times and that of abdominal obesity increased 1.032 times compared with when the stress rate was low. According to gender, for men, abdominal obesity (1.000 times) increased more than obesity (0.969 times), while, for women, obesity (1.168 times) increased more than abdominal obesity (1.072 times) (*p* < 0.01). In the case of alcohol consumption and smoking, the probability of obesity increased by 1.445 times and 1.194 times, respectively, compared with not drinking or smoking, while the probability of abdominal obesity increased by 1.291 times and 1.041 times, respectively. The increase in the probability of obesity due to alcohol consumption was 1.497 times in men and 0.782 times for women, whereas the increase in the probability of abdominal obesity was 1.439 times for men and 0.228 times for women. With regard to smoking, men showed a 1.272 times increase in the probability of obesity and a 1.086 times increase in the probability of abdominal obesity, whereas women showed a 0.847 times increase in the probability of obesity and a 0.835 times increase in the probability of abdominal obesity. Not practicing aerobic exercise increased the probability of obesity by 0.894 times and that of abdominal obesity by 1.025 times. The probability of obesity increased by 0.913 times in men and 0.832 times in women, whereas the probability of abdominal obesity increased by 1.002 and 1.336 times in men and women, respectively (*p* < 0.01) (Table 3).

## 4. Discussion and Conclusions

Even with the same BMI, health risks vary depending on the amount of visceral fat [16]. Abdominal obesity increases the risk of metabolic syndromes, including hypertension, diabetes, coronary artery disease, abnormal cholesterol levels, and nonalcoholic fatty liver disease [17].

According to the US Obesity Prevalence and Comorbidity Map of 2021, more than 42% of the US population, over 100 million people, accounting for 73.1% of the adult population, are obese or overweight [18]. In 1992, a survey of obesity prevalence in the United States found that 27% of women and 24% of men were obese [19]. However, according to the 2017–2018 National Health and Nutrition Examination Survey in Korea, there was little difference in the obesity rates between women (42.1%) and men (43%). Still, the severe obesity rate (BMI ≥ 40) was higher in women, at 11.5%, compared to 6.6% in men. In Europe, 59% of adults are reported to be overweight or obese. It is noteworthy that the obesity rate in men is increasing at a faster pace compared to women [20]. In this study, obesity was more prevalent in men, whereas abdominal obesity was more prevalent in women. Abdominal obesity serves as a risk signal for fat distribution and is a key factor in predicting increased risk of death [21].

The obesity rates in the United States and Europe have been increasing rapidly compared to Korea. However, Korea’s obesity level has remained relatively stable around 34% since 2015, with a slight decrease to 33.8% in 2019. Nevertheless, following the COVID-19 pandemic, there was a significant increase to 38.3% in 2020 [10].

Factors influencing female obesity include age, marital status, living arrangements, education, smoking [22], lifestyle habits [23], lack of exercise, high-calorie intake, anxiety, anger, depression, and weakened social support. Various factors, such as lifestyle habits, social and economic factors, and health-related behaviors and habits, affect obesity. Surveys conducted in various countries, including the United States, Australia, China, Switzerland, and Taiwan, through national nutrition surveys, have confirmed that occupation, education, economic status, social status, behavior patterns, and dietary patterns influence the occurrence of obesity [24,25,26]. This study classified the participants into those with and without obesity and abdominal obesity and examined the factors influencing obesity and abdominal obesity based on social characteristics and health behaviors. There were differences in influencing factors for increasing obesity and abdominal obesity depending on socioeconomic factors and health behavior factors. Therefore, this study suggests that obesity and abdominal obesity are obesity problems that must be considered and managed together.

Among the 5262 subjects in the study, the obesity rate was 38.3% (2014 individuals), and the abdominal obesity rate was 39.1% (2057 individuals). Abdominal obesity was more prevalent than obesity; men tended to have obesity, whereas women tended to have abdominal obesity.

Despite various studies worldwide, obesity remains a social issue as a health hazard, and the number of obese individuals continues to increase. The Korea National Health and Nutrition Examination Survey, which examines health statistics in Korea, is an annual nationwide survey. In the present study, a cross-sectional analysis was conducted using data from 2020, when social activities for the entire population were restricted because of COVID-19.

Several researchers have conducted studies to understand the impact of obesity on quality of life from various perspectives. However, unlike studies that focus on the impact of obesity on quality of life, this study divided the factors influencing obesity and abdominal obesity into socioeconomic and health behavior factors and analyzed the impact factors according to gender. This approach allows for the exploration of policy strategies and management of obesity rates at individual- and gender-specific levels. Additionally, various health management supports focusing on population groups with demographic factors related to obesity risk should be provided through health education programs. This includes ensuring that everyone gets enough sleep and maintains healthy lifestyle habits such as exercise, nonsmoking, and moderate alcohol consumption. The limitation of this study is that the survey elements of national data are limited, so the analysis must be limited, and the influence of factors such as underlying disease or living environment cannot be analyzed. Therefore, this study suggests that, in order to analyze specific causality, it is necessary to select subjects according to the research purpose and conduct experimental research on specific factors.

## Figures and Tables

**Table 1 healthcare-12-00827-t001:** Obesity rates according to socioeconomic characteristics.

Characteristics			Total Number of Subjects (*n* = 5262)					Total (%)
			Nonobese (BMI < 25) (*n* = 3248, 61.7%)	Obese (BMI ≥ 25) (*n* = 2014, 38.3%)	Nonabdominal Obese (wc M < 90 W < 85) (*n* = 3205, 60.9%)	Abdominal Obese (wc M < 90 W < 85) (*n* = 2057, 39.1%)	*p*-Value	
Socioeconomic characteristics	Gender	Male	43.4	61.8	44.6	60.6	0.000	50.6
		Female	56.6	38.2	55.4	39.4		49.4
	Age group	Youth (20~39)	35.1	33.3	38.9	26.7	0.000	34.4
		Middle (40~59)	40.0	41.3	40.4	40.6		40.5
		Senior (over 60)	25.0	25.4	51.5	48.5		25.1
		Mean ± SD	47.33 ± 16.49	47.95 ± 15.52	45.48 ± 15.88	51.07 ± 15.93	0.000	47.57 ± 16.12
	Marital status	Married	73.5	75.9	71.1	80.1	0.000	74.4
		Single	26.5	24.1	28.9	19.9		25.6
	Region (administrative district)	Urban	85.3	84.3	85.9	83.2	0.000	84.9
		Rural	14.7	15.7	14.1	16.8		15.1
	Income quartile (family)	Low	13.3	13.7	11.9	16.1	0.000	13.5
		Low-middle	21.6	22.1	20.8	23.4		21.8
		Middle-high	29.3	30.9	31.0	28.2		30.0
		High	35.8	33.2	36.3	32.3		34.8
	Educational attainment	≤Elementary	11.1	11.5	8.8	15.4	0.000	11.2
		Middle school	7.7	7.9	6.6	9.8		7.8
		High school	38.1	37.7	38.7	36.7		38.0
		≥University	43.1	42.9	46.0	38.1		43.0

%: weighted percentages.

**Table 2 healthcare-12-00827-t002:** Rates of obesity by health behaviors.

Characteristics			Total Number of Subjects (*n* = 5788)					Total(%)
			Nonobese (BMI < 25) (*n* = 3248, 61.7%)	Obese (BMI ≥ 25) (*n* = 2014, 38.3%)	Nonabdominal Obese (wc M < 90 W < 85) (*n* = 3205, 60.9%)	Abdominal Obese (wc M < 90 W < 85) (*n* = 2057, 39.1%)	*p*-Value	
Health Behaviors	Subjective health status	Good	31.2	28.3	33.3	24.7	0.000	30.1
		Middle	53.5	50.5	52.6	51.9		52.4
		Bad	15.3	21.2	14.1	23.4		17.6
	Stress perception	Low	71.8	69.4	71.3	70.2	0.000	70.9
		High	28.2	30.6	28.7	29.8		29.1
	Alcohol consumption	Yes	91.5	92.3	92.6	90.5	0.000	91.8
		No	8.5	7.7	7.4	9.5		8.2
	Smoking status	Yes	46.2	42.7	45.6	43.4	0.000	44.7
		No	53.8	57.3	54.4	56.6		55.3
	Aerobic exercise	Yes	42.8	46.2	45.4	41.9	0.000	44.1
		No	57.2	53.8	54.6	58.1		55.9
Sleeping duration		More than seven hours	66.4	60.3	67.6	58.0	0.000	64.0
		Less than seven hours	33.6	39.7	32.4	42.0		36.0
		Mean ± SD	7.06 ± 1.37	6.91 ± 1.37	7.11 ± 1.35	6.83 ± 1.40	0.000	7.00 ± 1.38

%: weighted percentages.

**Table 3 healthcare-12-00827-t003:** Logistic regression analysis of factors affecting obesity and abdominal obesity.

Characteristics	Total		Male		Female	
	Obesity	Abdominal Obesity	Obesity	Abdominal Obesity	Obesity	Abdominal Obesity
	OR (95% CI)					
Sex (ref. women)	2.318 *** (2.311–2.326)	1.857 *** (1.851–1.862)				
Age	0.991 *** (0.990–0.991)	1.007 *** (1.007–1.007)	0.987 *** (0.987–0.988)	1.007 *** (1.007–1.008)	1.042 *** (1.042–1.043)	1.030 *** (1.029–1.030)
Age (ref. over 60)						
age 20–39	1.084 *** (1.076–1.091)	0.867 *** (0.861–0.873)	1.114 *** (1.106–1.123)	0.914 *** (0.907–0.920)	1.477 *** (1.445–1.510)	0.965 *** (0.944–0.987)
age 40–59	1.051 *** (1.047–1.056)	0.953 *** (0.949–0.958)	0.956 *** (0.952–0.960)	0.895 (0.891–0.899)	1.660 *** (1.635–1.686)	1.379 *** (1.358–1.400)
Marital Status (ref. single)	1.273 *** (1.269–1.277)	1.051 *** (1.048–1.054)	1.344 *** (1.339–1.348)	1.071 *** (1.067–1.075)	0.663 *** (0.657–0.668)	0.709 *** (0.703–0.715)
Region rural (ref. urban)	1.137 *** (1.134–1.140)	1.163 *** (1.160–1.166)	1.112 *** (1.109–1.115)	1.130 *** (1.127–1.133)	1.830 *** (1.813–1.847)	1.833 *** (1.816–1.850)
Income (ref. High)						
Low	1.053 *** (1.049–1.057)	0.783 *** (0.780–0.786)	1.063 *** (1.059–1.067)	0.773 *** (0.770–0.776)	0.900 *** (0.891–0.909)	0.769 *** (0.761–0.777)
Middle-low	1.063 *** (1.061–1.066)	1.075 *** (1.072–1.078)	1.073 *** (1.070–1.076)	1.101 *** (1.098–1.105)	1.041 *** (1.032–1.049)	0.993 * (0.985–1.001)
Middle-high	1.178 *** (1.175–1.181)	1.016 *** (1.014–1.019)	1.194 *** (1.191–1.197)	1.026 *** (1.023–1.028)	1.018 *** (1.010–1.027)	0.939 (0.931–0.946)
Education (ref. university)						
≤Elementary	0.724 *** (0.721–0.728)	0.805 *** (0.802–0.809)	0.745 *** (0.741–0.749)	0.785 *** (0.781–0.789)	0.274 *** (0.270–0.279)	0.506 *** (0.498–0.514)
Middle school	0.847 *** (0.844–0.850)	1.033 *** (1.029–1.037)	0.725 *** (0.722–0.728)	0.870 *** (0.866–0.874)	1.418 *** (1.402–1.434)	2.220 *** (2.195–2.245)
High school	0.870 *** (0.868–0.872)	0.999 (0.997–1.001)	0.823 *** (0.821–0.825)	0.953 *** (0.951–0.956)	1.627 *** (1.616–1.638)	1.668 *** (1.657–1.680)
Sleep problem (ref. more than seven hours)	1.310 *** (1.305–1.314)	0.919 *** (0.918–0.920)	1.304 *** (1.299–1.308)	1.076 *** (1.072–1.080)	1.538 *** (1.523–1.553)	1.478 *** (1.463–1.492)
Subjective Health level (ref. Good)						
Middle	0.702 *** (0.699–0.704)	0.522 *** (0.520–0.523)	0.616 *** (0.614–0.618)	0.464 *** (0.462–0.466)	1.002 (0.992–1.012)	0.738 *** (0.731–0.746)
Poor	0.723 *** (0.721–0.725)	0.662 *** (0.660–0.664)	0.628 *** (0.626–0.630)	0.595 *** (0.593–0.597)	1.538 *** (1.525–1.551)	1.056 *** (1.048–1.065)
Current depression (ref. no)	1.217 *** (1.211–1.223)	1.030 *** (1.055–1.065)	0.882 *** (0.876–0.888)	0.986 *** (0.980–0.993)	1.506 *** (1.493–1.519)	1.072 *** (1.063–1.082)
Stress perception (ref. low)	1.020 *** (1.017–1.022)	1.032 *** (1.030–1.035)	0.969 *** (0.966–0.971)	1.000 (0.998–1.003)	1.168 *** (1.160–1.175)	1.082 *** (1.075–1.089)
Alcohol consumption (ref. no)	1.445 *** (1.433–1.458)	1.291 *** (1.281–1.302)	1.497 *** (1.484–1.511)	1.438 *** (1.426–1.451)	0.782 *** (0.755–0.809)	0.228 *** (0.220–0.236)
Smoking status (ref. no)	1.194 *** (1.192–1.197)	1.041 *** (1.039–1.044)	1.272 *** (1.269–1.275)	1.086 *** (1.083–1.088)	0.847 *** (0.842–0.853)	0.835 *** (0.830–0.841)
Aerobic exercise (ref. yes)	0.894 *** (0.892–0.896)	1.025 *** (1.023–1.027)	0.913 *** (0.911–0.915)	1.002 ** (1.000–1.005)	0.832 *** (0.827–0.837)	1.336 *** (1.328–1.345)
wald χ^2^	18,4251.280	302,565.429	42,084.011	115,596.724	3,733,360.497	438.765
Nagelkerke	0.058	0.056	0.049	0.039	0.165	0.139

Note: * *p* < 0.1, ** *p* < 0.05, *** *p* < 0.01.

## Data Availability

Data are contained within the article. The analyzed data are available from the author upon reasonable request.

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
