# Peer review of "Factors Affecting Abdominal Obesity: Analyzing National Data"

_healthcare, 2024, doi:10.3390/healthcare12080827_

Round 1
Reviewer 1 Report
Comments and Suggestions for Authors
Accept with minor revisions
Suggested revisions:
Affiliation #1 is incomplete, email is not listed
Abstract
No comments
Introduction
Lines 32-33: “Obesity is easily measurable through height and weight as key indicators of health.” This statement is false and needs to be replaced with factual and accurate information. Height and weight themselves are not key indicators of health. There is mounting pressure to remove BMI as an indicator of health. Waist circumference is a better indicator of obesity, and there is a wealth of literature to support this (as mentioned in your methods section). Please either update or remove this sentence.
Lines 56-58: “However, 2020 brought a global emergency, i.e., the COVID-19 pandemic, which caused an anticipated increase in obesity rates due to changes such as increased time spent at home and reduced social activities.” This statement needs to re-worded with factual information. There is an issue with the wording “anticipated increase in obesity rates”, this is also untrue. There were no anticipations or expectations that obesity rates would increase, just as there was no realistic expectations made on how much time populations would be facing lockdown measures preventing them regularly enjoyed activities pre-pandemic. If increases in obesity were anticipated, the suggested measures would have been different to prevent further health risks. It seems that the pandemic provided an opportunity to study obesity determinants after-the-fact based on the provided references.
Lines 69-71: “The goal was to provide a foundational rationale for preventive and management strategies to increase the obese and abdominally obese populations, considering various approaches and priorities.” This statement is contradictory; goal is to prevent obesity, but also manage strategies to increase the obese and abdominally obese populations. This needs to be re-worded to clearly state the goal of the research.
Methods
Line 103: change “old age” to senior to keep wording for the 60 years+ age group consistent.
Line 104: explain the 4 quartiles of household income, include currency values
Results
Lines 164/192/262: Tables need formatting improvements, and the titles do not need <>.
Discussion and Conclusions
Line 285: factors influencing female obesity were discussed. This section needs to be expanded to include factors influencing male obesity and compare/contrast the differences or similarities between sexes.
Acknowledgements/ Funding/ Author Contributions
These sections are missing from the manuscript.
References
There are major formatting adjustments needed between references 15-44. Additionally, a strange symbol appears with the journal name.
Author Response
We sincerely appreciate your thorough review and feedback. We have revised the manuscript as your review for submission. Thank you for your diligent efforts in reviewing.
|
Backmatter |
Thank you for reviewing the study. We added the backmatter incorporating the points you mentioned. Funding
Informed Consent Statement
Data Availability Statement
Conflicts of Interest Author Contributions Contribution of the authors can be summarized in following manner. Conceptualization: GH.K.; Formal analysis: HK.W., YA.J.; Supervision: GH.K.; Visualization: YA.J; Writing—draft: GH.K.,HK.W.,YA.J.; Writing—review and editing: GH.K.,HK.W.,YA.J. |
|
Reviewer1 |
|
|
Introduction Lines 32-33: “Obesity is easily measurable through height and weight as key indicators of health.” This statement is false and needs to be replaced with factual and accurate information. Height and weight themselves are not key indicators of health. There is mounting pressure to remove BMI as an indicator of health. Waist circumference is a better indicator of obesity, and there is a wealth of literature to support this (as mentioned in your methods section). Please either update or remove this sentence.
|
Thank you for the insightful review. We indeed proceeded with your suggestion to delete the sentence as you mentioned. |
|
Lines 56-58: “However, 2020 brought a global emergency, i.e., the COVID-19 pandemic, which caused an anticipated increase in obesity rates due to changes such as increased time spent at home and reduced social activities.” This statement needs to re-worded with factual information. There is an issue with the wording “anticipated increase in obesity rates”, this is also untrue. There were no anticipations or expectations that obesity rates would increase, just as there was no realistic expectations made on how much time populations would be facing lockdown measures preventing them regularly enjoyed activities pre-pandemic. If increases in obesity were anticipated, the suggested measures would have been different to prevent further health risks. It seems that the pandemic provided an opportunity to study obesity determinants after-the-fact based on the provided references.
|
Thank you for your feedback. We found and responded to your feedback by finding a paper that supports this, added the factual information. However, the obesity rate remained relatively stable around 34% after an increase from 31.7% in 2007 to 33.2% in 2015, then reaching 33.8% in 2019. However, there was a significant increase to 38.3% in 2020, and as of 2022, it has maintained at 37.2% (Ministry of Health and Welfare, 2023). We added the reference:
|
|
Lines 69-71: “The goal was to provide a foundational rationale for preventive and management strategies to increase the obese and abdominally obese populations, considering various approaches and priorities.” This statement is contradictory; goal is to prevent obesity, but also manage strategies to increase the obese and abdominally obese populations. This needs to be re-worded to clearly state the goal of the research. |
Thank you for the thorough review and valuable feedback. We made the necessary revisions to the sentences as you suggested. |
|
Methods Line 103: change “old age” to senior to keep wording for the 60 years+ age group consistent.
|
Thank you very much for reviewing the paper thoroughly, even down to the smallest details. We revised the text to incorporate the changes you suggested, including replacing the word "old age" with "senior" as per your review. |
|
Line 104: explain the 4 quartiles of household income, include currency values |
Thank you for the thorough review and valuable feedback. We added the sentence. The goal was to provide basic evidence for prevention and management strategies for the obese and abdominal obesity population, taking into account different approaches and priorities. |
|
Results Lines 164/192/262: Tables need formatting improvements, and the titles do not need <>. |
Thank you for the meticulous review. As per your suggestion, We improved the formatting of the tables and removed the <>. |
|
Discussion and Conclusions Line 285: factors influencing female obesity were discussed. This section needs to be expanded to include factors influencing male obesity and compare/contrast the differences or similarities between sexes. |
Thank you for your review. we revised the statement as follows: "In Europe, 59% of adults are reported to be overweight or obese. It is noteworthy that the obesity rate in men is increasing at a faster pace compared to women." We added the reference: Eurostate. European health interview survey Overweight and obesity - BMI statistics. 2024.
|
|
Acknowledgements/ Funding/ Author Contributions These sections are missing from the manuscript. |
Thank you for reviewing the study. We added the backmatter incorporating the points you mentioned. Funding
Informed Consent Statement
Data Availability Statement
Conflicts of Interest Author Contributions Contribution of the authors can be summarized in following manner. Conceptualization: GH.K.; Formal analysis: HK.W., YA.J.; Supervision: GH.K.; Visualization: YA.J; Writing—draft: GH.K.,HK.W.,YA.J.; Writing—review and editing: GH.K.,HK.W.,YA.J. |
|
References There are major formatting adjustments needed between references 15-44. Additionally, a strange symbol appears with the journal name. |
Thank you for the valuable review. We reviewed and made adjustments to the formatting of all references as you suggested. |

Reviewer 2 Report
Comments and Suggestions for Authors
There are many concerns with grammar and sentence structure throughout the paper. For instance, the first sentence of the paper should read more like, "Obesity is a global health problem with an increasing risk in all nationality, gender, and age groups." And the third sentence seems unnecessary ("Obesity is easily measurable through height and weight as key indicators of health.")
And the sentence structure of the sentence that starts on line 102 is confusing. And the sentence after that sentence needs clarification.
Some sentences are too long and complex, such as the sentence that starts on line 108.
The sentence that starts on line 132 is confusing. I think it should read more like, "A cross-analysis was performed to examine the association between socioeconomic factors (gender, age, marital status, household income, residential area, and education level) and health-related characteristics (subjective health perception, sleep duration, perceived stress, depression, alcohol consumption, smoking, and aerobic exercise), and obesity and abdominal obesity among the study population. Weights were applied to compensate for population structure and missing values.
The sentence that starts on line 146 is confusing. The sentence on line 149 could read more like, "Significant age-related differences were observed, with higher obesity rates in middle-aged (41.3%) and young (33.3%) individuals, and higher abdominal obesity rates in older (48.5%) and middle-aged (40.6%) individuals."
Most of the results section is not clear. For instance, it's not clear as to which groups or categories are being compared in the sentence starting on line 169.
Comments on the Quality of English LanguageThere are many concerns with grammar and sentence structure throughout the paper. For instance, the first sentence of the paper should read more like, "Obesity is a global health problem with an increasing risk in all nationality, gender, and age groups." And the third sentence seems unnecessary ("Obesity is easily measurable through height and weight as key indicators of health.")
And the sentence structure of the sentence that starts on line 102 is confusing. And the sentence after that sentence needs clarification.
Some sentences are too long and complex, such as the sentence that starts on line 108.
The sentence that starts on line 132 is confusing. I think it should read more like, "A cross-analysis was performed to examine the association between socioeconomic factors (gender, age, marital status, household income, residential area, and education level) and health-related characteristics (subjective health perception, sleep duration, perceived stress, depression, alcohol consumption, smoking, and aerobic exercise), and obesity and abdominal obesity among the study population. Weights were applied to compensate for population structure and missing values.
The sentence that starts on line 146 is confusing. The sentence on line 149 could read more like, "Significant age-related differences were observed, with higher obesity rates in middle-aged (41.3%) and young (33.3%) individuals, and higher abdominal obesity rates in older (48.5%) and middle-aged (40.6%) individuals."
Most of the results section is not clear. For instance, it's not clear as to which groups or categories are being compared in the sentence starting on line 169.
Author Response
|
We sincerely appreciate your thorough review and feedback. We have revised the manuscript as your review for submission. Thank you for your diligent efforts in reviewing. |
|
There are many concerns with grammar and sentence structure throughout the paper. For instance, the first sentence of the paper should read more like, "Obesity is a global health problem with an increasing risk in all nationality, gender, and age groups." And the third sentence seems unnecessary ("Obesity is easily measurable through height and weight as key indicators of health.") |
Thank you for the thorough review and valuable feedback. We made the necessary revisions to the sentences as you suggested, and I have also removed the third sentence. |
|
And the sentence structure of the sentence that starts on line 102 is confusing. And the sentence after that sentence needs clarification. |
Thank you for your feedback. We revised this paragraph as follows.
Independent Variables Socioeconomic (gender, age, marital status, household income, and education level) and health-related (perceived stress, sleep duration, depression, smoking, alco-hol consumption, subjective health perception, and aerobic exercise) characteristics were selected as independent variables to investigate their relationship with obesity and abdominal obesity. Socioeconomic characteristics (gender, age, marital status, household income, education level) and health-related characteristics (perceived stress, sleep time, depression, smoking, drinking, subjective health perception, aerobic exer-cise) were selected as independent variables. Among the socioeconomic characteristics, adults aged 20 years and older were categorized into three groups (youth, middle age, and old age senior), at 20-year intervals. Among socioeconomic characteristics, adults aged 20 or older were classified into three groups (young, middle-aged, and senior) at 20-year intervals. Household income was categorized into the fourth quartile of household income. Household income was classified into quartiles of household income classified according to the table of sam-ple household income quintiles in the National Health and Nutrition Examination Survey data. Marital status was divided into married and unmarried, and educational level was classified as elementary school graduate or below, middle school graduate, high school graduate, or college graduate or above. Marital status was categorized as married and single, and educational background was categorized as elementary school graduation or less, middle school graduation, high school graduation, and college graduation or higher. |
|
Some sentences are too long and complex, such as the sentence that starts on line 108. |
Thank you for your feedback. We revised the sentences as follows.
Based on the 4th National Health Promotion Plan (2015-2020), smoking, drinking, physical activity, cognitive stress, sleep time, depression, and subjective factors were added. Through a literature review, sleep time, which is considered an important variable in health behavior, was divided into less than 7 hours and more than 7 hours (Oh Sun-myeong et al., 2008; Ra Jin-sook et al., 2014). Subjective health status, which had been classified into 5 categories (bad to very good) in the National Health and Nutrition Examination Survey data, was reclassified into 3 categories: good, average, and poor. Depression was classified as present only if diagnosed by a physician. The aerobic exercise practice rate was classified based on the periods of moderate-intensity physical activity, high-intensity physical activity, moderate-intensity physical activity, and high-intensity physical activity mixed according to the National Health and Nutrition Examination Survey data analysis guidelines.
|
|
The sentence that starts on line 132 is confusing. I think it should read more like, "A cross-analysis was performed to examine the association between socioeconomic factors (gender, age, marital status, household income, residential area, and education level) and health-related characteristics (subjective health perception, sleep duration, perceived stress, depression, alcohol consumption, smoking, and aerobic exercise), and obesity and abdominal obesity among the study population. Weights were applied to compensate for population structure and missing values. |
Thank you for the thorough review and valuable feedback. I have made the necessary revisions to the sentences as you suggested. |
|
The sentence that starts on line 146 is confusing. The sentence on line 149 could read more like, "Significant age-related differences were observed, with higher obesity rates in middle-aged (41.3%) and young (33.3%) individuals, and higher abdominal obesity rates in older (48.5%) and middle-aged (40.6%) individuals." |
Thank you for your feedback. We revised the sentences as you mentioned.
Significant age-related differences were observed, with higher obesity rates in middle-aged (41.3%) and young (33.3%) individuals, and higher abdominal obesity rates in older (48.5%) and middle-aged (40.6%) individuals. |
|
Most of the results section is not clear. For instance, it's not clear as to which groups or categories are being compared in the sentence starting on line 169. |
Thank you for review. We added the sentence as follow.
Results of comparing obesity and abdominal obesity according to health behavior characteristics |

Reviewer 3 Report
Comments and Suggestions for Authors
Introduction:
· Clearly describe General and Specific objectives of the study
· Acknowledge the work of other researchers on the topic , identify the gaps in knowledge which your study will fill
Methods:
This section is weak, needs improvement.
· Study design:
briefly describe the study design
· Study population:
Describe target population, study population and study population frame
Results:
· Result section is too long with two tables
· Summarize your results in one paragraph according to objectives of your study
Give your results in one table
Discussion:
· Discuss main findings of your study
· Compare results of your study with similar studies conducted elsewhere
Conclusion:
· Help reader to understand why your research matter to them
· Restate your study problem and why it’s important
· Summarize your major points and make your arguments clear
· State limitations of your study and their implications
Comments on the Quality of English Languageno comments
Author Response
|
Introduction: · Clearly describe General and Specific objectives of the study Acknowledge the work of other researchers on the topic , identify the gaps in knowledge which your study will fill |
Thank you for your feedback. We revised the sentences as follows. However, 2020 brought a global emergency, i.e., the COVID-19 pandemic, which caused an anticipated increase in obesity rates due to changes such as increased time spent at home and reduced social activities. However, the obesity rate remained relatively stable around 34% after an increase from 31.7% in 2007 to 33.2% in 2015, then reaching 33.8% in 2019. However, there was a significant increase to 38.3% in 2020, and as of 2022, it has maintained at 37.2% (Ministry of Health and Welfare, 2023). The goal was to provide a foundational rationale for preventive and management strategies to increase the obese and abdominally obese populations, considering various approaches and priorities. The goal was to provide basic evidence for prevention and management strategies for the obese and abdominal obesity population, taking into account different approaches and priorities. |
|
Methods: · Study design:
· Study population: |
Thank you for review. We added the sentence as follow.
The methods section is structured into three main components: 1. Study Population, 2. Research Tools, and 3. Analysis Method, and below are the detailed sentences for each item to clearly describe them:
Since infants and adolescents are both a growth period and a school age period, the reasons for obesity will be different from those of adults and the intervention methods must also be different, so this study targeted adults. |
|
Results: · Result section is too long with two tables. Summarize your results in one paragraph according to objectives of your study |
Thank you for your review. But this study should be presented in two tables because they compared obesity and abdominal obesity by dividing them into socioeconomic factors and health behaviors. |
|
Discussion: · Discuss main findings of your study · Compare results of your study with similar studies conducted elsewhere |
Thank you for the excellent review. We appreciate your suggestions, and we have revised the Discussion section accordingly, incorporating the modifications you proposed.
According to the U.S. Obesity Prevalence and Comorbidity Map of 2021, more than 42% of the U.S. population, over 100 million people, accounting for 73.1% of the adult population, are obese or overweight (University of Chicago Opinion Research Center, NORC, 2023). In 1992, a survey of obesity prevalence in the United States found that 27% of women and 24% of men were obese (Kuczmarski, 1994). However, according to the 2017–2018 National Health and Nutrition Examination Survey in Korea, there was little difference in the obesity rates between women (42.1%) and men (43%). Still, the severe obesity rate (BMI ≥ 40) was higher in women, with 11.5%, compared to 6.6% in men. In Europe, over 52.7% of the adult population aged 18 and older is overweight, with little difference in obesity rates between women and men. However, the prevalence of overweight increases with age, particularly among men (Eurostat, 2019). In Europe, 59% of adults are reported to be overweight or obese. It is noteworthy that the obesity rate in men is increasing at a faster pace compared to women(Eurostat, 2024). In this study, obesity was more prevalent in men, whereas abdominal obesity was more prevalent in women. Abdominal obesity serves as a risk signal for fat distribution and is a key factor in predicting an increased risk of death (T. Pischon et al., 2008). The obesity rates in the United States and Europe have been increasing rapidly compared to those in Korea. However, Korea's obesity level has remained relatively stable around 34% since 2015, with a slight decrease to 33.8% in 2019. Nevertheless, following the COVID-19 pandemic, there was a significant increase to 38.3% in 2020 (Ministry of Health and Welfare, 2023).
|
|
Conclusion: · Help reader to understand why your research matter to them.
· Summarize your major points and make your arguments clear
·
State limitations of your study and their implications |
Thank you for the insightful review. We added sentences and revised this paragraph as follows.
The obesity rates in the United States and Europe have been increasing rapidly compared to those in Korea. However, Korea's obesity level has remained relatively stable around 34% since 2015, with a slight decrease to 33.8% in 2019. Nevertheless, following the COVID-19 pandemic, there was a significant increase to 38.3% in 2020 (Ministry of Health and Welfare, 2023).
This study classified the participants into those with and without obesity and abdominal obesity, and examined the factors influencing obesity and abdominal obesity based on social characteristics and health behaviors. There were differences in influencing factors for the increasing obesity and abdominal obesity depending on socieconomic factors and health behavior factors. Therefore this study suggest that Obesity and abdominal obesity are obesity problems that must be considered and managed together.
The limitation of this study is that the survey elements of national data are limited, so the analysis must be limited, and the influence of factors such as underlying disease or living environment cannot be analyzed. Therefore, this study suggests that in order to analyze specific causality, it is necessary to select subjects according to the research purpose and conduct experimental research on specific factors. |
|
Comments on the Quality of English Language. no comments |
We appreciate your thorough review of each section of the paper and the positive feedback provided. |
|
Change the title to "Obesity and Abdominal Obesity" - half your paper is about obesity in general |
Thank you for your review. We accepted your feedback and revised the title.
Factors Affecting Abdominal Obesity and Abdominal Obesity: Analyzing National Data |

Reviewer 4 Report
Comments and Suggestions for Authors
Change the title to "Obesity and Abdominal Obesity" - half your paper is about obesity in general
THE AUTHORS START THEIR STUDY DISCUSSING RESULTS OF OBESITY STUDIES LARGELY FROM THE U.S. GIVEN THIS LINE OF REASONING THEY SHOULD DISCUSS IN THE DISCUSSION SECTION WHETHER THE RESULTS FROM KOREA OR ASIA ARE SIMILAR OR DISSIMILAR TO THOSE IN THE WESTERN WORLD. IF THERE IS NO DIFFERENCE BETWEEN EAST AND WEST WHAT THEN WAS THE PURPSOE OF THIS STUDY? SHOULD A MULTIVARIATE EVALUATION HAVE BEEN DONE TO IDENTIFY THE MOST IMPORTANT / INDEPENDENT RISK FACTORS FOR CENTRAL OBESITY?
THE ARGUMENTS AND CONCLUSIONS ARE CONSISTENT. THERE IS HOWEVER NOTHING NEW.
ONE LAST POINT - THE AUTHORS WRITE"GETTING ENOUGH SLEEP." CHANGE THIS TO "OBTAINING ENOUGH SLEEP."
Author Response
|
We sincerely appreciate your thorough review and feedback. We have revised the manuscript as your review for submission. Thank you for your diligent efforts in reviewing. |
|
Review4 |
|
|
THE AUTHORS START THEIR STUDY DISCUSSING RESULTS OF OBESITY STUDIES LARGELY FROM THE U.S. GIVEN THIS LINE OF REASONING THEY SHOULD DISCUSS IN THE DISCUSSION SECTION WHETHER THE RESULTS FROM KOREA OR ASIA ARE SIMILAR OR DISSIMILAR TO THOSE IN THE WESTERN WORLD. IF THERE IS NO DIFFERENCE BETWEEN EAST AND WEST WHAT THEN WAS THE PURPSOE OF THIS STUDY? SHOULD A MULTIVARIATE EVALUATION HAVE BEEN DONE TO IDENTIFY THE MOST IMPORTANT / INDEPENDENT RISK FACTORS FOR CENTRAL OBESITY? |
Thank you for the insightful review. We discussed and added the sentence in the discussion section as follows.
This study classified the participants into those with and without obesity and abdominal obesity, and examined the factors influencing obesity and abdominal obesity based on social characteristics and health behaviors. Among the 5,262 subjects in the study, the obesity rate was 38.3% (2,014 individuals) and the abdominal obesity rate was 39.1% (2,057 individuals). Abdominal obesity was more prevalent than obesity; men tended to have obesity, whereas women tended to have abdominal obesity. |
|
THE ARGUMENTS AND CONCLUSIONS ARE CONSISTENT. THERE IS HOWEVER NOTHING NEW.
|
Thank you for the review. The most noble content we have found is as follows, and this sentence has been newly added during the revision. The obesity rate remained relatively stable around 34% after an increase from 31.7% in 2007 to 33.2% in 2015, then reaching 33.8% in 2019. However, there was a significant increase to 38.3% in 2020, and as of 2022, it has maintained at 37.2% (Ministry of Health and Welfare, 2023). |
|
ONE LAST POINT - THE AUTHORS WRITE"GETTING ENOUGH SLEEP." CHANGE THIS TO "OBTAINING ENOUGH SLEEP." |
Thank you for the thorough review and valuable feedback. I have made the necessary revisions to the sentences as you suggested. |
